

# Intact tactile anisotropy despite altered hand perception in complex regional pain syndrome: rethinking the role of the primary sensory cortex in tactile and perceptual dysfunction

Annika Reinersmann[1,3], Ian W. Skinner[2], Thomas Lücke[1], Nicola Massy-Westropp[3], Henrik Rudolf[4], G. Lorimer Moseley[2,3] and Tasha R. Stanton[2,3]

[1] Neuropediatric Department, Children's University Hospital St. Josef, Ruhr-Universität Bochum, Bochum, North-Rhine Westphalia, Germany
[2] Neuroscience Research Australia, NEURA, University of New South Wales, Sydney, New South Wales, Australia
[3] School of Health Sciences, University of South Australia, Adelaide, South Australia, Australia
[4] Department of Medical Informatics, Biometry and Epidemiology, Ruhr-Universität Bochum, Bochum, Germany

Corresponding author
Annika Reinersmann,
annika.reinersmann@rub.de

## ABSTRACT

Complex Regional Pain Syndrome (CRPS) is characterised by pain, autonomic, sensory and motor abnormalities. It is associated with changes in the primary somatosensory cortex (S1 representation), reductions in tactile sensitivity (tested by two-point discrimination), and alterations in perceived hand size or shape (hand perception). The frequent co-occurrence of these three phenomena has led to the assumption that S1 changes underlie tactile sensitivity and perceptual disturbances. However, studies underpinning such a presumed relationship use tactile sensitivity paradigms that involve the processing of both non-spatial and spatial cues. Here, we used a task that evaluates anisotropy (i.e., orientation-dependency; a feature of peripheral and S1 representation) to interrogate spatial processing of tactile input in CRPS and its relation to hand perception. People with upper limb CRPS ($n = 14$) and controls with ($n = 15$) or without pain ($n = 19$) judged tactile distances between stimuli-pairs applied across and along the back of either hand to provide measures of tactile anisotropy. Hand perception was evaluated using a visual scaling task and questionnaires. Data were analysed with generalised estimating equations. Contrary to our hypotheses, tactile anisotropy was bilaterally preserved in CRPS, and the magnitude of anisotropic perception bias was comparable between groups. Hand perception was distorted in CRPS but not related to the magnitude of anisotropy or bias. Our results suggest against impairments in spatial processing of tactile input, and by implication S1 representation, as the cause of distorted hand perception in CRPS. Further work is warranted to elucidate the mechanisms of somatosensory dysfunction and distorted hand perception in CRPS.

## INTRODUCTION

Complex regional pain syndrome (CRPS) is a relatively uncommon, multifactorial condition usually triggered by musculoskeletal or neural injury to a limb and characterised by pain, autonomic, sensory and motor abnormalities (*Birklein, O'Neill & Schlereth, 2015*; *Harden et al., 2010*; *Maihofner, Seifert & Markovic, 2010*; *Marinus et al., 2011*). In CRPS of the upper limb, there is evidence that primary motor (*Pleger et al., 2014*; *Schwenkreis et al., 2003*) and primary somatosensory cortex (S1) representation of both hands (hereafter called S1 hand representation) differ from those of controls and response profiles for tactile input are altered (*Di Pietro et al., 2015*; *Enax-Krumova et al., 2017*; *Lenz et al., 2011*; *Pleger et al., 2004*). There are also abnormalities in tactile sensitivity (i.e., ability to detect change in spatial and/or non-spatial tactile cues; see Table 1), namely tested by two-point discrimination (TPD), as well as abnormalities in the perceived hand size or shape (hereafter called hand perception) (*Lewis & McCabe, 2010*; *Lewis & Schweinhardt, 2012*; *McCabe et al., 2005*; *Moseley, 2004*; *Peltz et al., 2011*). Together this has led to the assumption that impairments in tactile sensitivity and hand perception can be attributed to alterations in S1 hand representation (*Lewis & Schweinhardt, 2012*; *Peltz et al., 2011*; *Pleger et al., 2005*; *Pleger et al., 2004*; *Schwenkreis, Maier & Tegenthoff, 2009*).

There are problems however, reconciling such assumptions with the full body of literature, particularly given inconclusive research findings, recent neuroimaging findings, and limitations of previously used tactile sensitivity measures proposed to evaluate spatial features of tactile processing(e.g., TPD). Firstly, studies that relate assessments of conscious hand perception with TPD-thresholds provide inconclusive results (*Lewis & Schweinhardt, 2012*; *Moseley & Wiech, 2009*; *Peltz et al., 2011*; *Schwenkreis, Maier & Tegenthoff, 2009*) and no studies have directly assessed the inter-relationship between hand perception, S1 hand representation and TPD-thresholds. Secondly, neuroimaging findings in CRPS show alterations in S1 hand representation for both hemispheres rather than only the affected hemisphere (*Di Pietro et al., 2015*; *Di Pietro et al., 2016*; *Lenz et al., 2011*), and fine-grained primary somatotopic maps appear normal in people with CRPS (*Mancini et al., 2019*), even though tactile sensitivity dysfunction and hand perception abnormalities appear confined to the affected limb. Finally, most studies that purport to evaluate spatial processing of tactile input actually use measures—most commonly TPD threshold measures—that require judgments based on both non-spatial (i.e., intensity, vibration or temporal) cues and spatial (i.e., orientation, width/ length) cues (*Bruns et al., 2014*; *Cashin & McAuley, 2017*; *Catley et al., 2014*; *Craig & Johnson, 2000*; *Green, 1982*; *Stanton et al., 2013*; *Todd, 2012*; *Tong, Mao & Goldreich, 2013*). Therefore, attributing deficits in such tests to impaired processing of tactile spatial cues depends on an assumption that has not been verified. As such, it remains unclear whether tactile sensitivity dysfunction in CRPS is due to impaired processing of non-spatial cues, spatial cues, or both.

**Table 1  Somatosensory information processing in tactile modality: Terms, types, task, and measure.**

| Tactile sensitivity | Type of response | Type of somato-sensory cue | Type of task | | Type of measure |
|---|---|---|---|---|---|
| Level at which tactile input exceeds the receptor threshold and elicits a neural response | Neural activity with somato-sensory correlate | | Orientation[a] | 2 point-orientation (2PD) Groove orientation (GO) | Acuity: the smallest detectable difference in spatial properties of tactile stimuli |
| | | | Width/length[a] | Tactile distance judgment (TDJ) | Accuracy: correctness of response to detection of spatial properties of tactile stimuli |
| Degree at which tactile stimulation is registered consciously | Conscious registration of sensation called a percept | Spatial cue | Comparison of width/lengths within a body part[a] | TDJ comparison based on stimuli orientation | Anisotropy: perceiving an across stimulus pair as farther apart than an along stimulus pair |
| Resolution, nature and capacity of the system to accurately extract information of a spatial or mechanical nature about a tactile stimulus | Acuity of percept (smallest detectable difference) Bias of percept (tendency to perceive equal stimuli as different) | | | | Anisotropic Bias: perceiving an across stimulus pair as farther apart than an along stimulus pair despite equal distances |
| | | | | | Magnitude refers to the degree to which an across stimulus pair is perceived as farther apart than an along stimuli, when actual stimuli differences do exist (e.g., despite across actually being closer) |
| | Accuracy of percept (correctness of response) | | | | Accuracy: correctness of response to spatial comparison of tactile distances |
| | | Non-spatial cue, spatial cue | Pressure/intensity | 2 point discrimination (TPD) | Accuracy: correctness of response to detection of mechanical (+/-spatial) properties of tactile stimuli |
| | | | Vibration/texture | Smooth -grooved discrimination Gap detection task | Acuity: the smallest detectable difference in mechanical (+/-spatial) properties of tactile stimuli |

**Notes.**

This table elaborates on terms of somatosensory information processing in the tactile modality that are used throughout the paper. Types of somatosensory information processess, the cues and responses which are used in tasks to measure processing tactile input are given.

[a]Evaluates spatial resolution using body representation as a frame of reference.
Alternative tactile paradigms exist (see Table 1), such as two-point-orientation discrimination (2POD) or grating orientation (GO), that are thought to better reflect spatial features of tactile processing and S1 functional morphology than TPD (*Adamczyk et al., 2016*; *Bruns et al., 2014*; *Craig & Johnson, 2000*; *Longo & Haggard, 2011*; *Tong, Mao & Goldreich, 2013*). These alternative paradigms incorporate the orientation-dependent nature that characterises spatial processing of tactile input, a property referred to as anisotropy (*Craig & Kisner, 1998*; *Gibson & Craig, 2002*; *Gibson & Craig, 2005*; *Knight Fle, Longo & Bremner, 2014*; *Longo & Haggard, 2011*; *Todd, 2012*; *Tong, Mao & Goldreich, 2013*; *Weber, 1996*).

Anisotropy is a base feature of our somatosensory system (also present in vision) (*Gibson & Craig, 2005*; *Wong & Chiang Price, 2014*), wherein across-oriented stimuli (aligned perpendicular to the axis of the limb) are perceived as farther apart than along-oriented stimuli (aligned along the axis of the limb) (*Longo & Haggard, 2011*; *Weber, 1996*). Tactile anisotropy is thought to result from orientation-selective neurons in S1 and the tactile receptive field size and shape on the skin's surface. It is unclear how such central and peripheral anisotropies interact (*Tong, Mao & Goldreich, 2013*; *Wong & Chiang Price, 2014*) and link to the perceptual bias that identical distances feel longer in the across- than in the along-orientation (anisotropic perception bias) (*Longo, Azanon & Haggard, 2010*; *Longo & Haggard, 2012*; *Miller, Longo & Saygin, 2016*). Even less understood is the process by which both tactile anisotropy and tactile homuncular representation in S1 (e.g., wide, 'fat' hand representation with higher numbers of receptive fields for across- versus along- the hand) are integrated to result in our *conscious* perception of a hand's physical size, which is typically not consciously perceived as wide and 'fat' (*Longo, 2015*; *Miller, Longo & Saygin, 2016*). That previous investigation of tactile sensitivity in CRPS has not isolated spatial from non-spatial cues implies that the assumption of dysfunctional spatial processing of tactile input in CRPS remains to be established. Further, such knowledge then suggests that we know little of the true relationship between hand perception and spatial processing of tactile input.

We aimed to fill this gap by investigating tactile anisotropy in people with CRPS, in people with non-CRPS hand pain and in pain-free controls and exploring the relationship between assessments of tactile anisotropy (i.e., a measure specific to spatial cues) and hand perception. Here we used a tactile distance judgement task whereby participants were touched on the skin in two spatial locations (via two wooden posts; termed stimulus pair), and in two sequentially distinct presentations based on orientation (along and across the dorsum of the hand). Participants then judged in which stimulus pair the wooden posts felt farther apart—those applied across or those applied along the dorsum of their hand. When the across and along stimulus pairs are of the same distance (e.g., both have wooden posts that are 3 cm apart), anisotropic perception bias is evaluated. Regardless of distance, to make such a judgement, participants must evaluate the spatial resolution of each stimulus pair (distance estimation) using their hand representation as a frame of reference and then compare these two separate distance estimations. Assessing anisotropy, therefore, provides a unique opportunity to capture spatial processing of tactile input in the context of a specific body part. Tactile distance judgements are particularly relevant to
evaluate given recent work showing that distortions in tactile distance perception at the hand are mirrored in the neural representation in S1 (*Tamè et al., 2019*).

We hypothesized that anisotropy (both tactile anisotropy and anisotropic perception bias) would be larger for the affected hand in people with CRPS than for the unaffected hand, and for either hand of non-CRPS pain and pain-free controls, representing disrupted spatial processing of tactile input. Further, we hypothesised that this magnified anisotropy and anisotropic perception bias would be positively related to disrupted hand perception. Because treatments targeting hand perception and S1 hand representation have both shown promising analgesic effects in CRPS (*Longo, 2015*; *Moseley & Wiech, 2009*), the need to further clarify the interaction between hand perception, spatial processing of tactile input and S1 hand representation appears justified (*Pleger et al., 2005*).

## METHOD

### Participants

The convenience sample included people with Complex Regional Pain Syndrome (CRPS), controls with pain of other origin (PC), and controls without pain (healthy subjects). This study was approved by the University of South Australia (UniSA) Human Research Ethics Committee (No. 35944). All recruited participants provided written informed consent and the study was performed in accordance with the ethical standards of the Declaration of Helsinki (1991).

Recruitment of all groups occurred at three sites: the School of Health Sciences, UniSA; Royal Melbourne Hospital Rehabilitation Hand Therapy Clinic; Neuroscience Research Australia, Sydney. Patients with pain were recruited using posters displayed at UniSA and in clinical settings, through recruitment appeals on social media or within existing research databases, and via word of mouth. Healthy participants were recruited via institutional students and members of the staff, through public advertisement and included pain participants' relatives, if interested.

Participants were screened via email or telephone and prior to participation, clinically screened by a trained health professional on the day of testing to assess symptoms/signs and function. Patients with CRPS type I were required to have received an original diagnosis of the condition by a pain specialist and to meet the CRPS research criteria on the day of testing (see Table 2 in results section for frequency of sensory, sudomotor, vasomotor, trophic, and motor signs). PC participants were required to have pain in the upper limb for at least 6 months. Healthy participants were required to be pain-free in the upper limb and have no history of trauma, or medication intake (aside of oral contraceptives). Potential participants, who reported a current diagnosis of a neurological or psychiatric condition, were excluded.

An *a priori* sample size was calculated using G*power software (Version 3.1). We initially powered this study to detect a small to medium effect ($f = 0.2$) using a 3 (Group) $\times$ 2 (affected versus unaffected limb) repeated measures analysis of variance (RM ANOVA), with 80% power, alpha of 0.05, correlation for repeated measures of 0.5, and adjusting for a 10% withdrawal rate. This resulted in a total of 45 participants (15 participants/group).

**Table 2  Clinical signs and symptoms in CRPS patients and patients with upper limb pain of other origin.**

| | CRPS I | | | Upper limb pain of other origin | | | p-values |
|---|---|---|---|---|---|---|---|
| | Left-affected (n) 9 | Right-affected (n) 6 | Total (n) 15 | Left-affected (n) 7 | Right-affected (n) 9 | Total (n) 16 | |
| Current pain (m ± sd) | 4.3 ± 2.3 | 4.50 ± 2.5 | 4.4 ± 2.3 | 4.9 ± 2.1 | 3.5 ± 2.1 | 4.1 ± 2.1 | p = 0.086 |
| Average pain (m ± sd, past 4 weeks) | 5.3 ± 2.1 | 5.5 ± 2.6 | 5.4 ± 2.3 | 3.7 ± 1.7 | 2.6 ± 1.8 | 3.1 ± 1.8 | p = 0.110 |
| illness duration (m ± sd, months) | 58.1 ± 104.7 | 52.2 ± 35.2 | 55.7 ± 81 | 62.7 ± 57.9 | 87.7 ± 62.4 | 76.8 ± 59.9 | p = 0.460 |
| DASH (m ± sd, Disability of Hand/Shoulder/Arm) | 69.3 ± 19.2 | 89.2 ± 11.4 | 77.3[*] ± 18.9 | 47.1 ± 14.5 | 52. 5 ± 24.2 | 50.0 ± 19.7 | p = 0.001 |
| Neglect-like severity score (m ± sd) | 1.1 ± 0.9 | 1.8 ± 0.9 | 1.4[*] ± 0.9 | 0.6 ± 0.8 | 0.4 ± 0.6 | 0.5 ± 0.7 | p = 0.005 |
| Bath Body perception scale (m ± sd) | 17.6 ± 0.9 | 20.6 ± 6.3 | 18.8[*] ± 5.7 | 12.6 ± 6.9 | 12.7 ± 7.9 | 12.6 ± 7.1 | p = 0.014 |
| Range of motion (ROM; Ratio) | 0.6 ± 0.3 | 0.66 ± 38 | 0.58[*] ± 0.33 | 0.87 ± 0.18 | 0.79 ± 0.26 | 0.83 ± 0.21 | p = .0012 |
| Hyperalgesia (e.g., tactile/bland pressure) | 5 | 5 | 10[*] | 0 | 0 | 0 | p = .004 |
| Hypaesthesia (tactile) | 5 | 5 | 10[*] | 0 | 0 | 0 | p = .000 |
| Dynamic (touch) allodynia | 4 | 2 | 6[*] | 1 | 1 | 2 | p = .008 |
| Vasomotor | 6 | 5 | 11[*] | 0 | 2 | 2 | p = .001 |
| Trophic | 6 | 5 | 11[*] | 0 | 0 | 0 | p = .000 |
| Sudomotor | 8 | 5 | 13[*] | 2 | 1 | 3 | p = .000 |

**Notes.**

CRPS 1, Complex regional pain syndrome type 1; SD, standard deviation; NRS, numerical rating scale; DASH, disability of the hand, arm and shoulder instrument; ROM, range of motion.

*Significant difference between CRPS I and upper limb pain of other origin (ANOVA), analysis of variance or $\chi^2$.

Given that tactile distance judgements have never before been evaluated in people with CRPS, we were unable to base this effect size directly on past studies. However, we purposefully chose a small to medium effect size because past work evaluating TPD threshold in people with CRPS suggested that powering for such effects was reasonable. Specifically, differences in TPD threshold between CRPS and healthy volunteers were of a moderate to large effect (Cohen's $d = 0.63$) and TPD threshold differences between the unaffected and affected limb in CRPS were of a moderate effect (Cohen's $d = 0.53$) (*Lewis & Schweinhardt, 2012*) Our choice was also informed by guidelines for meaningful effect sizes in highly heterogenic clinical populations (*Gignac & Szodorai, 2016*; *Lipsey et*

*al., 2012*). Following consultation with a biostatistician, generalised estimating equations (GEE) were chosen to allow a more comprehensive analysis of the data (e.g., to allow consideration of other clinical factors that might influence tactile distance judgements). While the ability to undertake formal power analyses based on GEE is limited, importantly, GEE has been shown to have higher power with lower sample sizes and/or lower numbers of repeated measures than RM ANOVAs (*Ma, Mazumdar & Memtsoudis, 2012*), suggesting we would be adequately powered.

## Questionnaires and assessments
### Assessment of handedness and clinical symptoms and function of the affect limb

Handedness was assessed in all participants with a standardized inventory (*Oldfield, 1971*). Participants with chronic pain of either CRPS or other origin additionally completed questionnaires that assessed pain, pain duration and function of the affected limb. Pain (current and average pain level over the past 4 weeks) was rated on an 11-point pain numerical rating scale (0 indicating no pain at all, 10 indicating worst pain imaginable). Pain duration was computed in weeks. A structured questionnaire was used to assess the presence of sensory, vasomotor, sudomotor and motor symptoms. Function of the affected limb was assessed using the Disability of Hand Shoulder and Arm (DASH) questionnaire (*Beaton et al., 2001*) an openly accessible self-report instrument of 30 items assessing physical functioning. The DASH delivers a mean score ranging from 0 (no function) to 100 (full function).

### Assessment of hand perception

Participants with chronic pain of either CRPS or other origin also completed measures of body representation that assessed the perception of the affected hand. The presence of foreign limb feelings (FLF) were assessed with the questionnaires Bath CRPS Body Perception Disturbance (BPD) (*Lewis & McCabe, 2010*) and the Neglect-like questionnaire (*Frettloh, Huppe & Maier, 2006*; *Galer & Jensen, 1999*) for which higher scores denote greater disturbance. We also used a depictive body representation assessment, namely a template matching task (*Moseley, 2005*) in which a series of photographs of male or female hands were provided to participants. The photographs had a black background and were taken of the hand dorsum in a first-person perspective. All photographs were resized in four increments (demagnification: −5% and −10%; magnification: +15% and +30%, see example in Fig. S1) using a customised LabVIEW program. After each testing block (see below), participants were asked to identify the photograph (matched to their gender) that best corresponded to their subjective perception of the size of their hand.

### Assessment of clinical signs

In addition to the pain specialist's diagnosis and medical report, a trained clinician assessed clinical signs (sensory, vasomotor, sudomotor, and motor/trophic) with a standardized protocol. The Budapest research criteria (*Harden et al., 2010*) were used to determine presence of CRPS (i.e., presence of at least one symptom in all 4 categories and at least one sign in two categories). For sensory signs, hyperalgesia was assessed by using a pressure

algometer over the 3rd PIP joint of the middle finger and thenar eminence, brush-evoked allodynia was tested with 10 brush strokes at 1 Hz hand dorsum and tactile sensitivity with monofilaments. For vasomotor signs, skin colour asymmetries were assessed by observation, temperature asymmetries using an infrared tympanic thermometer. For sudomotor signs, oedema was evaluated by measuring the circumference at the wrist crease, the dorsum and proximal phalanx of middle finger; and signs of sweating were assessed by observation. For motor/trophic signs, resting and intentional tremor were assessed by observation, range of motion was measured using a goniometer for wrist and finger movements as well as visual evaluation of the ability to perform finger-thumb opposition. Signs of muscle weakness were assessed by testing grip strength with a dynamometer.

## PROCEDURES

### Tactile distance judgement

We used a recently developed anisotropy paradigm that involves judging tactile distances applied across and along the hand (*Longo & Haggard, 2011*), the area that is frequently reported as the centre area of perceptual distortion in CRPS patients (feels swollen or larger than actual physical size (*Lotze & Moseley, 2007*; *Moseley, 2005*; *Peltz et al., 2011*).

We used a bespoke device (Fig. 1): two wooden posts were mounted on a wooden disk (to assure stable stimulation) at three different distances: two, three or four centimetres (cm) apart. Here, we used three distance combinations (2/3; 2/4; 3/3), evaluating three length differences (0 cm/1 cm/2 cm). For testing this resulted in five distance combinations, based according on which stimulus-pair was provided across versus along the hand axis: (i) 2 across/3 along; (ii) 3 across/2 along; (iii) 2 across /4 along; (iv) 4 across/2 along; (v) 3/3—across/along. In the context of identical tactile distances (3 cm/3 cm), a stimulus presented across the hand dorsum typically feels wider (subsequently described as farther apart) than a stimulus with identically distanced wooden posts but presented along the hand dorsum. This constitutes a perceptual bias resulting from the spatial processing of tactile input and is proposed to mirror the morphology of S1 hand representation (*Knight Fle, Longo & Bremner, 2014*; *Longo, Ghosh & Yahya, 2015*; *Miller, Longo & Saygin, 2016*).

We followed the protocol presented by Longo and Haggard (*Longo & Haggard, 2011*) that has been used consistently in the literature (*Knight Fle, Longo & Bremner, 2014*; *Longo, Ghosh & Yahya, 2015*; *Miller, Longo & Saygin, 2016*). This validated protocol for tactile distance judgement testing (*Longo & Haggard, 2011*) involved participants sitting with eyes closed, arms and palms resting flat on a table, fingers splayed. The bespoke device with two wooden posts of varying distance combinations (2, 3, and 4 cm apart) was used to provide tactile stimuli to the hand (termed a stimulus pair). Two different tactile distance stimulus pairs were provided to the dorsum of the hand: one stimuli pair across the hand (perpendicular to the longitudinal axis of the limb) and one stimuli pair along the hand (parallel to the longitudinal axis of the limb). Participants were asked to judge which stimulus pair (of the two provided) felt farther apart. For example, if one stimulus pair consisting of rods a distance of 2 cm apart was delivered across the hand and one stimulus pair with a distance of 3 cm between the two rods was delivered along the hand,

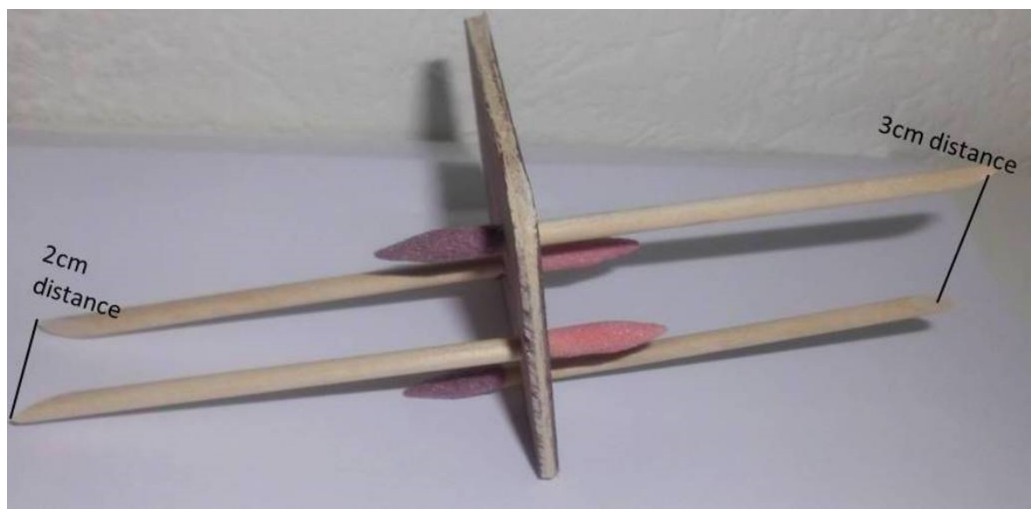

**Figure 1** **Tactile distance judgement disk.** Our bespoke tactile distance judgement disk. Each disk held two stimulus pairs; one on each side. Here, one side of the stimulus holds one stimulus pair where the two rods are mounted two centimetres (cm) apart on the disk and one stimulus pair where the two rods are mounted three cm apart.

participants underwent a forced choice paradigm, identifying which of the two stimuli pairs felt farther apart. Each stimulus pair application lasted for approximately one second with an interstimulus interval (ISI) of approximately one second.

Participants underwent one test session which consisted of two blocks (one block for each hand). During each block, each of the above outlined five distance combinations were presented on the back of the hand, either across the axis of the hand or along the axis of the hand. For each hand, 50 tactile distance judgements were made (5 distance combinations x 10 repetitions). The order of trials was pseudorandomized, and the tested hand order was counterbalanced. This was chosen to minimize attention trade-off and to reduce the risk of increased pain or pressure sensitivity in people with CRPS or hand pain.

## Tactile distance judgement outcomes

Two primary outcomes were evaluated to comprehensively investigate processing of spatial features of tactile input: anisotropic perception bias and tactile anisotropy (See Fig. 2).

### *Primary outcome one—anisotropic perception bias*

This bias was evaluated by considering the frequency of perceiving across distances as farther apart, despite equal stimulus pair length (*Longo & Haggard, 2011*; *Miller, Longo & Saygin, 2016*). That is, bias was operationalised as the proportion of 3 cm/3 cm stimulus pairs where the stimulus pair delivered across the back of the hand (across-oriented stimulus pair) was subjectively perceived as farther apart than the stimulus pair applied along the back of the hand (along-oriented stimulus pair). The presence of anisotropic perception bias is evidenced by >50% (i.e., greater than chance) of across-oriented stimulus-pairs being subjectively perceived as farther apart at this point of objective equality (POE, where width-difference between stimulus pairs equals 0: 3/3 stimulus pair). We predicted

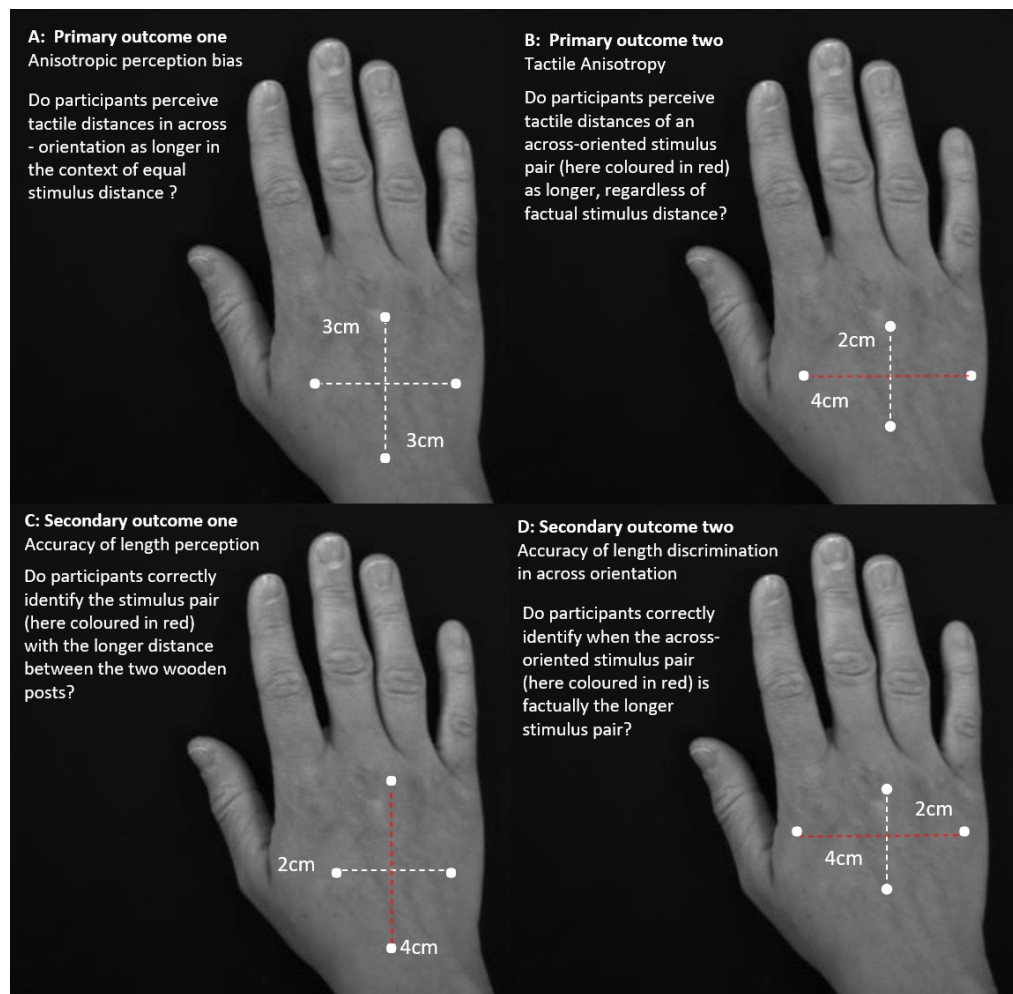

**Figure 2** **Primary and secondary outcome of response analyses.** Illustration of the two primary outcomes and the secondary outcomes according to which the responses were analysed. Primary outcomes one (A) and two (B) estimate tactile anisotropy, the perception of stimulus pair distances as longer in the across direction, dependent and independent of factual distance differences. Secondary outcome one (C) and two (D) evaluate response accuracy independent and dependent of stimulus pair orientation.

group differences in anisotropic perception bias for only the affected hand (e.g., a greater proportion of across-oriented stimuli perceived as farther apart for the CRPS affected hand, showing a perceptual overestimation of width).

### Primary outcome two—tactile anisotropy

This outcome evaluated the tendency to perceive the across-oriented stimulus pair as farther apart than the along-oriented stimulus pair, *regardless of factual width-difference between stimulus pairs* (i.e., to perceive a distance as farther apart independent of whether

it factually is farther apart) (*Craig & Kisner, 1998*; *Gibson & Craig, 2002*; *Gibson & Craig, 2005*; *Weber, 1996*). To this end, all stimulus pairs (2/3, 3/3, 2/4, 3/2, 4/2) were considered, with this outcome operationalised as the proportion of stimuli, where the stimulus pair delivered across the hand was perceived as farther apart than the stimulus pair applied along the hand. Using all stimulus pairs allows us to determine the point at which the subjective perception of stimulus pair width changes (exceeds chance: >50%), providing an indication of the magnitude of bias (see also Table 1 for terminology). We hypothesised increased tactile anisotropy on the CRPS affected hand (for the PSE to occur farther away from the POE).

Two secondary outcomes related to accuracy of tactile distance judgements were also evaluated (See Fig. 2). Specifically, these explored the participant's capacity to correctly discern when a stimulus pair is farther apart than the other stimulus pair (both independent of and dependent on the stimulus-orientation). This was completed to ensure that participants could correctly discern factual differences in stimuli when they were present. Thus, responses to stimulus distance combinations of 2 cm/3 cm and 2 cm/4 cm were used for these measures. Accuracy measures are important to place anisotropic perception bias and tactile anisotropy results in context. For example, an impairment in accuracy (i.e., the ability to correctly discern and represent the distance between two tactile points) would raise the possibility that any changes in anisotropy or bias might merely reflect group differences in other features that alter accuracy, such as tactile cutaneous innervation density or spacing.

### Secondary outcome one—accuracy of tactile distance perception (orientation-independent)

This outcome evaluated the ability to accurately detect a tactile width-difference between two tactile stimuli-pairs, when there was a factual difference, regardless of whether the farther apart stimulus pair was provided along or across the hand. That is, it merely evaluated the ability to perceive and correctly identify the farther apart stimulus pair provided. This outcome was operationalised as the proportion of stimulus pairs in which the distance that felt farther apart was correctly identified (using responses to stimulus distance combinations 2 cm/3 cm and 2 cm/4 cm). If this measure differed between groups, it would suggest that people with CRPS were merely inaccurate on all judgements.

### Secondary outcome two—accuracy of tactile distance perception (orientation-dependent)

This outcome evaluated the ability to accurately detect a tactile width-difference between two stimuli-pairs when the across-oriented stimuli pair was actually farther apart than the along-oriented stimuli pair (*Craig & Kisner, 1998*; *Gibson & Craig, 2005*). This measure explores accuracy of tactile distance judgements more comprehensively by focusing on accurate width-discrimination in the across-orientation on the hand (and for which a deficit could possibly be 'washed out' and missed when considering orientation *independent* accuracy) This outcome was operationalised as the proportion of responses of correct recognition of longer stimulus pair in the across-orientation only (using stimulus distance combinations of 2 cm/3 cm and 2 cm/4 cm). Similar to above, if this measure differed

between groups, it would support that people with CRPS were inaccurate overall in detecting across-stimuli that felt farther apart and would suggest any findings for anisotropic perception bias were invalid.

## Data analyses (using IBM SPSS Statistics 24.0)
### Clinical data and body perception
Differences between groups in baseline measures and body perception questionnaires were evaluated using separate analyses of variance (ANOVA). Differences between groups in hand perception, measured via the visual scale task, were compared using chi-squared analysis.

### Tactile distance judgements
Generalised estimating equation (GEE) models for binary data with logit link function (*Zeger & Liang, 1986*) were used to evaluated the effect of group, the effect of hand (affected/unaffected), and the effect of hand perception on the four tactile distance judgement outcomes (binary dependent variables; e.g., yes/no response for 'across stimuli pair judged farther apart than the along stimuli pair' for 3/3 cm stimulus pairs for anisotropic perception bias). Analysis using GEE was chosen to allow for investigation of the contribution of numerous different demographic and clinical features to tactile distance judgement performance. GEE follows the procedure of regressing variables of interest against the binary outcome variable of interest on a univariate and multivariate level. In backwards regression, the best model of fit for the data is then identified, using the QUICC (quasi likelihood under the independence) criterion as an indicator for model of fit (*Zeger & Liang, 1986*; *Zeger, Liang & Albert, 1988*).

For each of the four outcomes, the variable of Group as well as a set of predictor variables of interest for all participants (general predictor set), including hand (affected/non-affected hand for participants suffering from chronic pain and dominant/non-dominant hand for pain-free participants), age, gender, current pain and length of stimulus pair difference (0/1/2 cm), were entered on a univariate and on a multivariable level. Then, in participants with chronic pain (i.e., the two patient groups), the variable of Group (Pain of other origin as reference standard), a set of clinical explanatory variables (pain levels, illness duration, signs of sensory dysfunction, signs and symptoms of motor impairment), and the degree of hand perception disturbance were explored to clarify the role of sensory function and body perception in tactile distance judgements. All variables are listed with their beta/covariate values, 95% CI about the estimate and *p*-value for each outcome measure in the File S1.

### Primary analysis (anisotropic perception bias and tactile anisotropy)
This involved three stages. First, a descriptive analysis considering the frequency of stimulus pair responses was undertaken. That is, the frequency of across-stimuli perceived as farther apart exceeding chance (>50%) for 3/3 stimulus pairs, and for 2/3, 3/3, and 2/4 stimulus pairs, was considered to provide evidence for the presence of bias and anisotropy, respectively.

Second, the above-described GEE analysis was undertaken. This analysis provides an odds ratio (OR) for each group that represents the odds of that group perceiving touch

from an across-stimulus pair as farther apart than from an along-stimulus pair, relative to the pain-free group (i.e., the reference standard). Including clinical predictors, such as hand perception, allowed us to determine if the odds of choosing an across-stimulus pair as farther apart differ as a function of (impaired) hand perception.

Third, curve fitting to determine the Point of Subjective Equality (PSE) for tactile anisotropy was undertaken. An additional GEE model (using the tactile anisotropy outcome: dependent variable of binary data response yes/no to 'across stimulus pair judged farther than the along stimulus pair') was used to generate data for determination of the PSE curve. Specifically, this GEE included a two-way interaction term for 'Group' and 'hand' and a three-way interaction term for the variables 'width difference' (0 cm/,1 cm,2 cm) 'Group' and 'hand'. Thus, 12 coefficients were estimated (i.e., two coefficients for each group by hand combination) and used to plot six logistic curves for the range of 'width differences' (i.e., range of $-2$ from stimulus pair 2/4 to $+2$ from stimulus pair 4/2 ). Using this data, the negative value of the ratio of the two estimated coefficients ($-$[2 way interaction coefficient/3-way interaction coefficient] for each combination of hand and group) determines the inflection point of the logistic growth curve, which corresponds to 50% probability of choosing across as farther apart than along and thus, estimates the PSE. A PSE significantly less than zero indicates that touch on the hand felt farther apart for across stimulus pairs than along stimulus pairs; a PSE greater than zero indicates that touch on the hand felt closer together for across than along stimulus pairs. The corresponding standard errors were calculated from the GEEs robust estimate of the covariance matrix via the Delta method. Last, we interpolated the PSE for the width difference at which participants perceived the across –stimulus as wider in the context of equal distance between the two wooden posts in across and along orientation equality (deviation from point of objective equality which would here demarcate 0 cm distance difference).

### *Secondary analysis*
This involved performing the above described stage two GEE analysis on outcomes of accuracy (orientation dependent and independent).

## RESULTS
### Study sample
Fifty-three people participated. Upon clinical screening on the day of testing, three CRPS patients were excluded as they did not meet the CRPS research criteria. Two patients with CRPS reported increased pain levels during tactile stimulation such that testing was discontinued on the affected hand. Therefore, complete datasets of both hands were available for 48 participants (CRPS: $N = 14$; PC: $N = 15$; healthy subjects: $N = 19$).

All participants were right-handed, except for one female patient with CRPS and one female patient with rheumatoid arthritis. Both patient groups were predominantly comprised of females (CRPS12/14; PC: 13/15; healthy participants: 9/19). Healthy participants were younger than the patient groups (mean, sd and range: CRPS $= 47.6 \pm 12.6$; 27–56; PC $= 58.7 \pm 17.1$; 27–85; healthy subjects $= 36.7 \pm 14.5$; 17–66; ANOVA main effect F $(2,49) = 9.5$; $p = 0.001$). Of patients with PC, one had carpal tunnel syndrome,

two reported chronic pain following soft tissue injury, 10 had rheumatoid arthritis, and 2 had osteoarthritis. Table 2 outlines mean ratings and frequencies of sensory, autonomic and motor signs and symptoms of both patient groups.

### Pain intensity and duration

Current and average pain intensity reports and duration of pain did not differ between the two patient groups (Table 2). CRPS patients reported significantly higher physical impairment (DASH) than PC patients (ANOVA main effect $F(1,30) = 14.8$; $p = 0.001$).

### Body perception disturbance

The feeling of limb foreignness (FLF) differed between CRPS patients and PC patients and was higher for CRPS patients than for those with PC, expressed in a higher Neglect-like score (ANOVA main effect $F(1,30) = 9.1$, $p = 0.005$). Similarly, CRPS patients reported higher distortion of hand perception than PC participants, expressed in a higher score on the Bath Body perception Scale score (ANOVA main effect $F(1,29) = 6.8$; $p = 0.014$).

### Hand perception (visual scale task)

Table 3 lists the frequency of chosen hand images that participants identified as matching the perceived size of the own affected (dominant) or non-affected (non-dominant) hand. The chosen template corresponding to the affected hand for patients with CRPS, PC and pain-free participants differed significantly between the three groups ($\chi 2(8) = 16.04$, $p = 0.042$). A post-hoc analysis of the contingency table with Bonferroni corrected $p$-values showed that pain–free participants were less likely than the pain groups (CRPS and non-CRPS pain) to match their dominant hand perception to a hand template that depicted an increase of 30% ($\chi 2(8) = 8.98$, $p = 0.002$). No group differences were found regarding the non-affected/non-dominant hand ($\chi 2(8) = 7.46$, $p = 0.87$).

### Tactile distance judgements

Table 4 reports all proportions and percentages of across-oriented stimulus pairs perceived as farther apart on the skin. Results are structured to first report the overall group comparison and then list predictors that are specific for the patient- group comparison. Further details on the general predictor variable set and odds for each predicting variable for each outcome measure are listed in the supplementary section (Tables 1–3).

### Primary analysis: Anisotropic perception bias

Descriptive analyses of response frequency support a clear anisotropic perception bias on dorsal surfaces of both hands in all participants (see Table 4). That is, of all equal distance stimulus pairs (3/3 cm), 69.8% (335/480; 95% CI [65.7–73.9]%) were perceived as farther apart when delivered in the across-orientation on the dorsum of the affected/dominant hand. Similarly, on the non-affected/non-dominant hand, 64.6% (310/480; 95% CI [60.3–68.9]%) of all equal distance stimulus pairs (3/3 cm) were perceived as farther apart when delivered in the across-orientation. In all groups, occurrence of anisotropic perception bias significantly exceeds chance on both hands, (50%, for OR and CI see general set of predictors and supplementary section). The magnitude of anisotropic perception bias was

**Table 3** Visual matching task assessing hand perception: frequency and percentage of each chosen template per group and per hand.

| | | CRPS | | Pain of other origin | | Healthy participants | | All participants | |
|---|---|---|---|---|---|---|---|---|---|
| | | Affected hand (*N*/%) | Non-affected hand (*N*/%) | Affected hand (*N*/%) | Non-affected hand (*N*/%) | Dominant hand (*N*/%) | Non-dominant hand (*N*/%) | Affected/dominant hand (*N*/%) | Non-dominant/non-affected hand (*N*/%) |
| | Template 1 magnification 30% | 4/28.6 | 1/7.1 | 0/0 | 0/0.0 | 3/15.5 | 5/26.3 | 7/14.3 | 6/12.3 |
| | Template 2 magnification 15% | 1/7.1 | 3/21.4 | 0/0 | 2/13.4 | 2/10.5 | 4/21.1 | 3/6.1 | 9/18.4 |
| | Template 3 no resizing | 2/14.3 | 5/35.7 | 3/20 | 5/33.3 | 5/26.3 | 4/21.0 | 11/22.4 | 15/30.6 |
| Perceived hand size template | Template 4 demagnification 5% | 2/14.3 | 3/21.4 | 6/40 | 5/33.3 | 9/47.4 | 3/15.8 | 17/34.7 | 11/22.4 |
| | Template 5 demagnification 10% | 5/35.7 | 2/14.3. | 6/40 | 3 /20.0 | 0/0 | 3/15.8 | 11/22.4 | 8/16.3 |

not different between participants with CRPS, PC, and healthy participants (see Table 4 and Table S1).

## General set of predictors for all participants

Anisotropic perception bias did not differ between groups (CRPS = OR: 0.501; CI 95% [0.215–1.171], $p = 0.111$; PC = OR: 0.531; CI 95% [0.244–1.159], $p = 0.112$; healthy subjects = reference group) or hands (OR: 0.783, CI 95% [0.518–1.182], $p = 0.244$) when controlling for gender, pain intensity and age (see Table 1 of the supplementary section). In this multivariate analysis, the best model of fit (QUICC: 1193.54) to explain the variance in perceiving across-oriented stimulus pairs as longer in the context of equal stimulus pair lengths (3/3) only included gender, current pain intensity and age. And of these variables, only gender was significant, finding that male participants had 0.5 times the odds of having anisotropic perception bias than female participants (OR: 0.507; CI 95% [0.300–0.857], $p = 0.011$).

## Patient specific clinical variable set

Considering only the pain groups, results show that a combination of gender, presence of allodynia, and current pain intensity best predicts the anisotropic perception bias in the context of equal distance (3/3 cm) stimulus pairs (QUICC 727.009) on a multivariable level. As above, male gender significantly reduced the log odds for anisotropic perception bias (OR 0.609, CI 95% [0.376–0.986], $p = 0.044$). Current pain intensity was a potential predictor (mean [95% CI] bias = 1.012 [1.000–1.023], $p = 0.054$). Notably, hand perception did not predict anisotropic perception bias—it was not included in the multivariate model (did not add to model fit), and univariate analyses were not statistically significant (Body

**Table 4    Responses to tactile stimuli.**

| Analyses | CRPS | | Pain of other origin | | Healthy participants | | All participants | |
|---|---|---|---|---|---|---|---|---|
| | Affected hand | Non-affected hand | Affected hand | Non-affected hand | Dominant hand | Non-dominant hand | Affected/dominant hand | Non-affected/non-dominant hand |
| Primary outcome one: anisotropic perception bias with equal length difference 0 cm) | 87/140 62.1% | 101/140 72.1% | 109/150 72.7% | 86/150 57.3% | 139/190 73.2% | 123/190 64.7% | 335/480 69.8% | 310/480 64.6% |
| Primary outcome two: tactile anisotropy (length difference 0, 1, 2 cm) | 407/700 58.1% | 427/700 61% | 450/750 60% | 409/750 54.3% | 559/950 58.8% | 557/950 58.6% | 1,416/2,400 59.0% | 1,393/2,400 58.0% |
| Secondary outcome one: tactile accuracy [a]correctly identified length difference (1 cm/2 cm) | 408/560 72.9% | 430 /560 76.8% | 461/600 76.8% | 471/600 78.5% | 629/760 82.8% | 606/760 79.7% | 1,498/1,920 78.0% | 1507/1920 78.5% |
| Secondary outcome two: accuracy of tactile anisotropy [a]correctly identified length difference in across orientation (1 cm/2 cm) | 224/280 80.0% | 238/280 85.0% | 251/300 83.7% | 247/300 82.3% | 332/380 87.4% | 330/380 86.8% | 807/960 84.1% | 815/960 84.9% |

**Notes.**
[a]Correct answer: indicates the number of stimuli pairs that were correctly identified as being longer. In anisotropic perception bias (primary outcome one), 'correct answer' indicates the number of stimuli pairs where length difference was identified as longer in across direction in the context of equal stimulus length. For tactile anisotropy (primary outcome two), 'correct answer' indicates the number of stimuli pairs wherein the across stimulus is perceived as longer. For secondary outcome one, tactile accuracy, 'correct answer' indicates the number of stimuli pairs where length difference was correctly identified regardless of orientation. For accuracy of anisotropy (secondary outcome two), 'correct answer' indicates the number of stimuli pairs where the across stimuli was correctly identified as having the greater length difference. See also Fig. 2.

perception scale: OR: 1.01; 95% CI of 0.97–1.05, $p = 0.54$). Similarly, presence of sensory dysfunction did not predict anisotropic bias.

## Primary analysis: generalised Tactile anisotropy

Descriptive analysis of response frequency supports the presence of tactile anisotropy occurring bilaterally in all participants. That is, across-oriented stimulus pairs were perceived as further apart independent of whether there is a factual difference in width-distance between stimulus-pairs and there were no differences between hands ($\chi^2(1)$: 0.004, $p = 0.948$). The proportion of across-oriented stimulus pairs being perceived as further apart on each hand are listed in Table 4 and the general predictor set for tactile anisotropy to occur are listed in the supplementary section Table 2.

### General set of predictors for all participants

Tactile anisotropy did not differ as a function of hand (OR 1.040, 95%CI [0.883–1.226], $p = 0.637$), but some groups differences were present. In the multivariate model of best fit (which included gender, length difference of stimulus pairs, current and average pain intensity), the PC group had reduced odds of perceiving across-oriented stimulus pairs as further apart than pain-free controls (OR: 0.632, CI 95% [0.407–0.981], $p = 0.041$; see Table 2 in the supplementary section). People with CRPS had no difference in the odds of perceiving across-stimuli as farther apart than pain-free controls (OR: 0.623, 95% CI [0.362–1.074], $p = 0.088$). While approaching statistical significance, the wide confidence intervals suggest large variability of any effect.

### Patient specific clinical variable set

Considering both pain groups, results show that a combination of illness duration, average pain intensity, and stimulus pair distance difference (0/1/2 cm) best predicts the tactile anisotropy on a multivariable level. Similar to above, hand perception did not predict tactile anisotropy—it was not included in the multivariate model (did not add to model fit), and univariate analyses were not statistically significant (Body perception scale OR: 1.10; 95% CI of 0.809–1.495, $p = 0.544$).

To further explore the anisotropy data, the GEE models using group x hand and group x hand x stimulus pair length difference interactions were used to create psychophysical curves of tactile distance judgements on both hands for all three groups (see Fig. 3). Visual inspection suggests that the slope increase of the curve for the affected hand of the CRPS group may differ from that of the pain-free control group but this trend does not meet statistical significance. Table 5 provides the PSE values for tactile distance judgements as function of hand and group. Here, PSE represents the point at which the distances between across- and along-oriented stimulus pairs are perceived as subjectively equal. These PSE values are largely similar between groups (i.e., we did not detect a statistically significant difference between the groups).

### Secondary analyses: Accuracy of tactile distance perception (orientation independent and orientation dependent)

The GEEs evaluating the tactile distance accuracy outcomes confirmed that all groups were equally accurate in discerning the longer length in both directions on both the affected (dominant) and non-affected (non-dominant) hand. Importantly, neither clinical nor sensory signs, nor distorted hand perception, were found to influence the accuracy of these tactile distance judgements (neither orientation-dependent nor independent). See the supplementary tables for the proportion of accurately identified stimulus-pairs in both directions and the general and patient-specific predictors for accurate tactile distance perception set.

## DISCUSSION

We hypothesized that anisotropic perception bias and tactile anisotropy, measured with tactile distance judgments, would be abnormal in CRPS. More specifically, we hypothesized

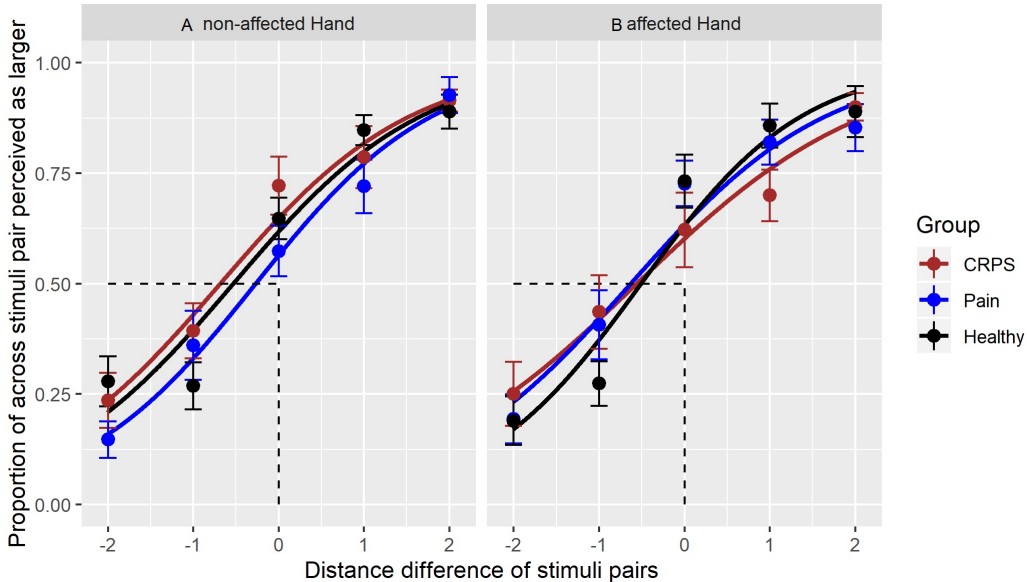

**Figure 3 Responses to tactile stimuli.** A logistic curve for each stimulus pair and the perceived proportion of stimuli perceived as farther apart in the across-orientation. Each range of 'width differences' (i.e., range of −2 from stimulus pair 2/4 to +2 from stimulus pair 4/2) is presented per group and per hand.The left part (A) shows the curve for the unaffected hand, the right part of the (B) the curve for the affected hand. The different curve colours represent the best-fit of responses for each group; the data-points show the means of the proportion of stimuli identified as farther apart, whereby a datapoint occurring at less than 0 indicates an anisotropic perception in the across-orientation. Error bars represent SE of the mean. The x-axis represents the stimulus—pair combinations with their distance differences: across-oriented 2 vs along-oriented 4 = −2, across-oriented 2 vs along-oriented 3 = −1, across-oriented 3 centimetre/along-oriented 3 centimetres = 0 (also point of objective equality), across-oriented 3/along-oriented 2 = 1, across-oriented 4/along-oriented 2 = 2.

**Table 5 Anisotropic perception bias, Point of subjective equality (PSE).** Shows the beta coefficients estimating the point of subjective equality (PSE), i.e., the point at which distances between across- and along-oriented stimulus pairs are perceived as subjectively equal. These PSE values are largely similar between groups that is, no statistically meaningful difference between groups was detected.

| Group | Hand | 2-way Interaction Group * Hand (estimated coefficents, $\beta$) | 3-way Interaction Group * Hand * Width (estimated coefficents, $\beta$) | Point of subjective Equality (PSE) |
|---|---|---|---|---|
| CRPS | Non-affected hand | 0.613 | 0.894 | −0.685 |
| | Affected hand | 0.412 | 0.739 | −0.561 |
| Pain of other origin | non-affected hand | 0.261 | 0.900 | −0.271 |
| | Affected hand | 0.551 | 0.962 | −0.631 |
| Pain-free | non-affected hand | 0.481 | 0.904 | −0.533 |
| | Affected hand | 0.540 | 1.058 | −0.510 |

that a magnified anisotropic perception bias on the affected in CRPS would be seen in CRPS and would be related to distortions in hand perception. Contrary to our hypotheses, anisotropic perception bias and tactile anisotropy were bilaterally preserved in people with CRPS and were comparable in magnitude to that seen in those with non-CRPS pain and in healthy participants. Additionally, neither tactile anisotropy nor anisotropic bias were related to measures of distorted hand perception in CRPS. Given these findings, it is relevant to review the contemporary body of knowledge on spatial processing of tactile input, composition of S1 hand representation and the possible role of cortical changes in CRPS.

That we did not detect a consistent group difference in the magnitude of anisotropic perception bias or in the presence of tactile anisotropy suggests that there is no compelling evidence for disrupted spatial processing of tactile input on the hand dorsum in CRPS. Given the link between tactile distance judgements and neural representation in S1, our findings also imply a preserved normative S1 hand representation in CRPS. Such a finding is unexpected given the well-described deficits in tactile sensitivity and hand perception, presumed to be related to abnormal S1 hand representations (*Catley et al., 2014*; *Lenz et al., 2011*; *Lewis & Schweinhardt, 2012*; *Maihofner et al., 2003*; *Peltz et al., 2011*; *Pleger et al., 2005*; *Pleger et al., 2004*).

## Anisotropic perception bias, and tactile anisotropy

One argument for the lack of magnified anisotropic perception bias (and tactile anisotropy) in people with CRPS is that they may merely be inaccurate at the task, and thus, consequently increase noise in the data renders a between-group signal undetectable. However, CRPS participants demonstrated correct recognition of width differences (independent of orientation; see supplementary analysis, see secondary outcome one), which suggests that our results were not due to discrimination errors. Bilaterally preserved accuracy—despite allodynia, pain or hypaesthesia—also suggests that receptive field and receptor properties on the hand's dorsum (and centrally) are not affected by CRPS pathophysiology. Similarly, that we did not detect a difference in anisotropy between hands in those with unilateral CRPS or unilateral hand pain (PC) suggests against a role of dysfunctional spatial processing of tactile input in contributing to hypoaesthetic signs or distorted hand perception, both of which were observed in the affected, but not unaffected, hands.

These results would not be predicted based on what is currently understood regarding the S1 representation of the affected and non-affected hand in CRPS or the contemporary knowledge of composition of S1 hand representation (*Azanōn et al., 2016*; *Di Pietro et al., 2015*; *Di Pietro et al., 2016*; *Gallace & Spence, 2010*; *Gallagher, 2005*; *Medina & Coslett, 2010*; *Serino & Haggard, 2010*). That is, given previous observed between-hand differences in S1 hand representation in unilateral CRPS, we would expect anisotropic differences as well. Here it is relevant to consider the underlying proposed mechanisms of anisotropy. While it is unclear how central and peripheral anisotropies are linked, tactile anisotropy is thought to constitute a general operating principle of body representation (*Green, 1982*; *Knight Fle, Longo & Bremner, 2014*; *Longo, Ghosh & Yahya, 2015*; *Miller, Longo & Saygin, 2016*).

For S1 hand representation, it is thought that the numbers of peripherally stimulated oval-shaped RFs are 'counted' cortically, and since the number of stimulated oval-shaped RFs on the hand's dorsum is higher in medio-lateral compared to proximo-distal orientation, limbs are represented 'wider and squatter' than they really are (*Longo & Haggard, 2011*). It is thought that this 'fat' bias of orientation-specific distance perception mirrors homuncular distortion of S1 hand representation (and thus may influence perceptions and cognitions of the hand) (*Longo & Haggard, 2011*; *Longo & Haggard, 2012*; *Miller, Longo & Saygin, 2016*).

Drawing on the proposed mechanisms underlying tactile anisotropy, it may therefore be possible that our results differ from past work because S1 morphology is actually disrupted in a different way than currently presumed for CRPS. Two limitations of the neuroimaging evidence for altered S1 representation in CRPS are relevant here. First, most studies are compromised by poor blinding, incomplete reporting or lack of control group or control hand (*Di Pietro et al., 2013*) which suggests caution when interpreting many CRPS neuroimaging results. Second, all studies have relied on Euclidean distance between thumb/index finger, and the fifth finger, which does not take into account that the cortex is curved, not flat (*Baliki et al., 2011*). A recent study that overcame this limitation demonstrated no abnormalities in digit-specific organisation of S1 in people with CRPS (*Mancini et al., 2019*) which seems consistent with the current findings and point to a lesser, or different, role of S1 changes in CRPS than has been widely accepted.

Notwithstanding challenges in imaging, it is certainly possible that tactile anisotropy might be preserved given that processing of other types of somatosensory input have been shown to be intact in CRPS. Indeed, while our results were surprising, past work has demonstrated that aspects of somatosensory processing and multisensory integration are preserved in people with CRPS—i.e., functions that involve neural mechanisms distinct from those of static TPD measures (*Moseley & Wiech, 2009*; *Reinersmann et al., 2013*; *Reiswich et al., 2012*). Such findings suggest that the somatosensory processes underlying measures of tactile distance judgements and TPD differ and are consistent with suggestions that TPD measures likely reflect a deficit of processing intensity cues rather than spatial cues in CRPS or a disruption in fine-grained maps of digits on the CRPS-affected hand (*Bruns et al., 2014*; *Craig & Johnson, 2000*; *Craig & Kisner, 1998*; *Mancini et al., 2013*; *Tong, Mao & Goldreich, 2013*).

## Distorted hand perception despite intact somatosensory spatial processing of tactile input

While spatial processing of tactile input was preserved, hand perception was distorted in CRPS and unrelated to anisotropic perception bias (i.e., the presumed indicator of the 'fat' S1 hand representation). Specifically, the outcome on the task measuring hand representation (tactile distance judgements) was unrelated to all measures of hand perception (visual and cognitive-affective hand perception, measured with template—matching and questionnaire).

Our results are generally consistent with both the lack of correlation between different body representation outcome measures (*Longo, 2015*) and the observed distorted

hand perception in CRPS (*Frettloh, Huppe & Maier, 2006*; *Galer & Jensen, 1999*; *Lewis & McCabe, 2010*). Body representation research often finds that outcomes of different measures are unrelated and suggests that this reflects the continuum along which the body is represented in different dimensions (somatosensory, visual, cognitive/affective etc.). On this continuum, top-down and bottom-up composed representations reference onto each other for any given task outcome (*Gallagher, 2005*; *Longo, 2015*; *Longo & Haggard, 2012*). The complexity of this interaction or reciprocal referencing may then also explain deficits in some types of body representation while other representations (such as S1 hand representation) are intact even within the same sensory modality (i.e., tactile).

This notion of complex representation is also relevant to the composition process of the percept itself: tactile stimulation starts a complex process of sensation that typically ends with a percept—one *feels* the touch at a specific location on the body. The percept results from bottom-up processing (tactile sensation) and top-down interpretation of these sensations against a stored inner S1 hand representation. A percept thus integrates a multiplicity of factors: receptor density in the skin, S1 receptor field geometry and orientation-selective neurons of receptor fields, cortical magnification and perceived hand size. Moreover, there are multifactorial influences on every type of body representation, whereby top-down interpretations also incorporate psychosocial & environmental imprints and reflect 'mentalisation' of sensorimotor signals (*Fotopoulou & Tsakiris, 2017*). Given all these proposed contributions to a percept, it is possible that some aspects are more heavily weighted in certain tasks than others and thus may well result in differences between body-representation outcomes. The ultimate percept may then simply reflect the reciprocal referencing of body representations that varies as a function of task demand or other factors.

## Relationship between S1 hand representation and hand perception

Despite the fact that tactile anisotropy exists ('fat, squat' S1 hand representation, (*Longo & Haggard, 2011*), we generally do not explicitly perceive our hands as fat and squat. It is intriguing that the phenomenology of CRPS distorted hand size perception appears to parallel what would be predicted with tactile input—and its anisotropic property—alone. That is, in healthy controls and people with non-CRPS hand pain, tactile distance perception is integrated into our conscious hand perception in a way that results in an accurate sense of the size and shape of our hand. However, in CRPS, tactile distance perception is normal and conscious hand perception is not. Is it possible that, in CRPS, the usual weighting of tactile distance perception and S1 hand representation is disrupted (see Fig. 4) and that the perceptual distortion reflects altered weighting of somatosensory input in the production of conscious hand perception?

This speculative idea has some backing in the literature. For example, in anorexia nervosa, there is greater than usual connectivity within somatosensory cortex and less than usual activity within areas associated with the visual processing of faces and bodies (extrastriate body area and fusiform area) (*Kim et al., 2017*) . This has led to the proposal that body perception disturbances in that group reflect the relatively greater weighting of somatosensory (tactile) body representations (i.e., wider, squatter) (*Longo, 2015*).

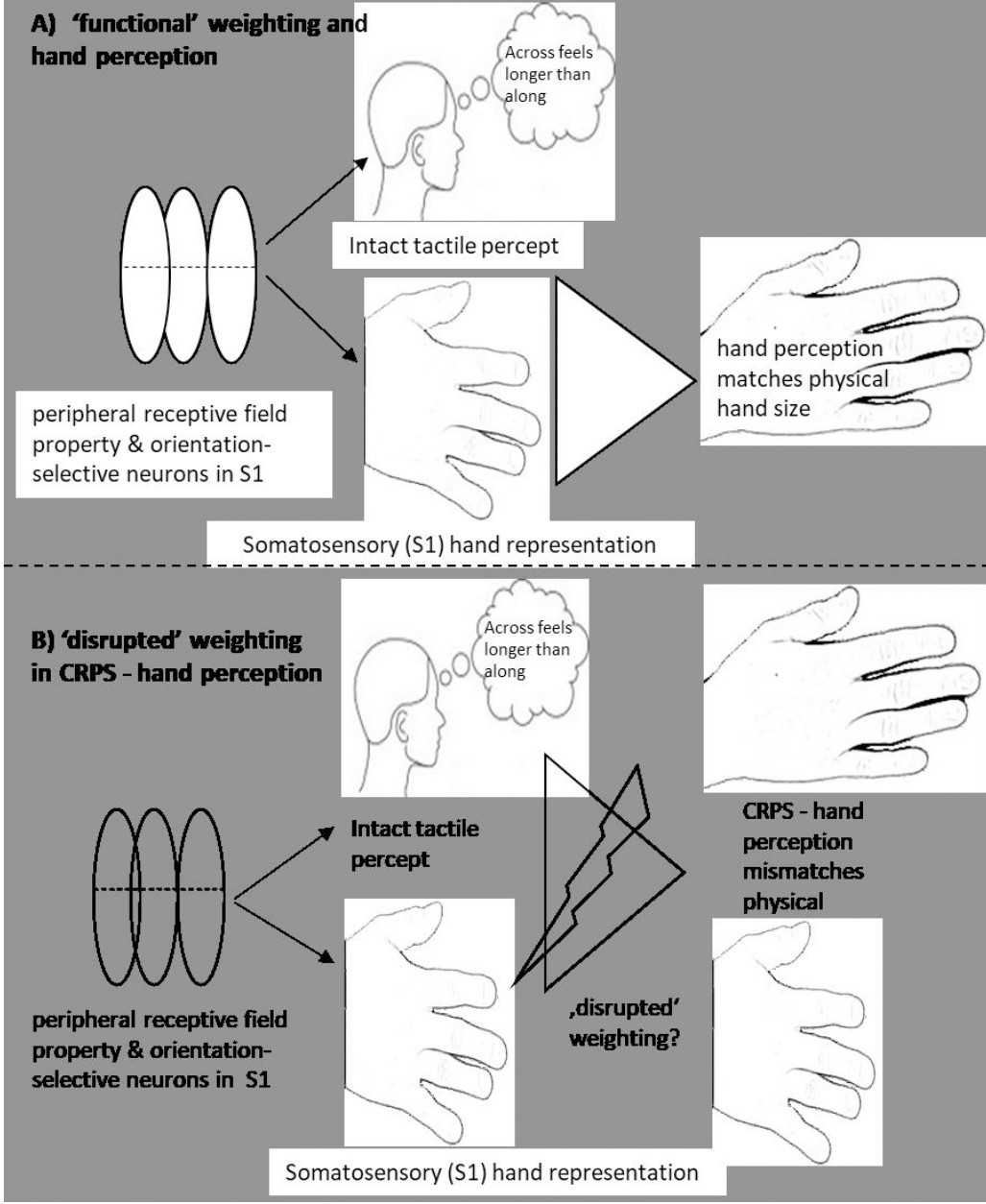

**Figure 4** **Implicit weighting process of different types of body representation.** An implicit weighting process and its resulting hand perception. In (A), the presumed 'functional' implicit weighting process is depicted where despite a homuncular distorted S1 hand representation the resulting conscious hand perception matches the physical hand size. In (B), a proposed 'altered' implicit translation process is depicted. This 'altered' implicit translation process might result in a distorted hand perception that is mirrored in the wide and squat hand perception of CRPS patients.

Furthermore, although an increased weighting of somatosensory representation might not conclusively explain anorectic perceptions (*Green, 1982*), 'swollen or fat' perceptions have been observed under conditions of neuronal or sensorimotor disruptions. For example,

local anaesthetic to the finger results in a perception of a 'fat' finger, with proprioceptive processing in S1 appearing to have a role in this distortion (*Gandevia & Phegan, 1999*; *Lackner, 1988*; *Moseley et al., 2006*). Indeed, mismatches between proprioception and other sensory/motor inputs have repeatedly been discussed in relation to altered CRPS body size/shape perceptions—and results of the present hand questionnaires match these discussions (*Lackner, 1988*; *Lewis et al., 2010*; *Moseley, 2004*; *Moseley et al., 2006*; *Reinersmann et al., 2010*; *Reinersmann et al., 2012*).

Together such findings suggest that while spatial processing of tactile input and S1 hand representation appear preserved (as our results suggest), disruptions in other sensory processing networks might be present and account for a 'altered weighting process'. One possibility is that altered or increased functional connectivity of somatosensory and motor cortex (S1/M1 sensorimotor cortex) to the right insular and other motor areas might impact on a weighting process of somatosensory regions—in terms of amplified weighing—and that this might contribute or result in an 'bizarre and foreign' hand perception. Imaging studies for example show alterations in functional connectivity between regions of the motor and intraparietal sulcus (IPS) that serve as an interface when integrating perceptual and motor information (*Bolwerk, Seifert & Maihofner, 2013*; *Kim et al., 2017*; *Maihofner, Handwerker & Birklein, 2006*; *Pleger et al., 2014*). Additionally, there is evidence in CRPS of reduced resting state functional connectivity between S1 and insula regions which show decreased endogenous inhibitory control of sensory processing regions in the insula (*Kim et al., 2017*).

Finally, we acknowledge that just as in anorectic body perception the distorted CRPS hand perception may reflect an increased weighting of top-down (cognitive-affective) interpretations rather than a (cortically driven) over-weighting of somatosensory (tactile) hand representation. In the context of aversive and ambiguous sensation (inexplicable pain, reduced motor function, decreased awareness of hand position, swollen feelings) and an unaesthetic appearance, the CRPS sufferer may perceive the hand as foreign and disturbing. There are some self-report data that suggest this to be the case (*Lewis & Schweinhardt, 2012*; *Michal et al., 2017*; *Moseley & Butler, 2017*).

Experiencing symptoms that even a pain specialist cannot explain may reinforce rejection of the hand or difficulties to integrate it into the own body concept. This in turn opens the option that psychoeducational interventions, and/or a combination of psychoeducational measures and cognitive behavioral therapy might impact on the functionality of mutually operating S1 hand representations. There is some preliminary support for this idea with observational data clearly suggesting better impacts on bodily perception from a psychoeducational approach tailored to CRPS than from a similar untailored approach (*Moseley & Butler, 2017*).

While this is the first study to use robust measures of spatial processing of tactile input in people with CRPS, and concurrently measure hand perception, a limitation of our study is that we did not include a neuroimaging component which would have allowed formal exploration of concurrent cortical changes. Future research linking neural and perceptual measures of the hand with evaluation of numerous features of spatial acuity is clearly warranted to delineate tactile dysfunction in people with CRPS. A second limitation is

that we did not formally lodge a detailed protocol prior to data collection or analysis, a procedure now recommended in pain studies to promote accountability, transparency and openness (*Lee et al., 2018*). We did develop an *a priori* experimental protocol that followed a standardized procedure in the use of the new paradigm and use of well-established questionnaires, as described in the methods section. Additionally, our *a priori* sample size calculation was powered to detect a small to moderate effect for the interaction between Group and Hand using a planned RM ANOVA. We instead used GEE to analyse the data, meaning our calculation was not specific to our analysis; but, as mentioned GEE analyses tend to have higher power with lower sample numbers than RM ANOVAs (*Ma, Mazumdar & Memtsoudis, 2012*), which suggests this may not have been a problem. It is relevant to note that the confidence intervals for tactile distance judgements in the CRPS group were very wide, suggesting that any between-group difference is likely small. This may suggest that our initial choice of powering for a small-to-moderate effect size may be too liberal. Regardless, even if significant differences were present between groups, our findings for anisotropic perception bias and tactile anisotropy were in the opposite direction than hypothesized. Last, our hand perception task used standardized hand images, meaning that between group differences in perceived hand size could reflect between group differences in actual hand size. However, individual differences in actual limb size are typically greater than group differences, suggesting against a systematic difference underlying the present results. That this task has been used in this way clinically and in pilot trials in our group, and between group differences between those with CRPS and pain-matched controls, supports its use here (*Moseley, 2005*). It does remain possible that group differences represent not a difference in perceived hand size but rather differences in the ability of an individual to transpose their own limb to the pictured limbs, but even then, we would expect increased unsystematic error. That body perception questionnaire responses about limb size typically matched the visual scaling task choices suggests that it does capture perceived hand size.

Our study did also have important strengths. These include the use of a paradigm that incorporates the orientation-dependence in spatial processing of tactile input, the use of a non-CRPS pain patient control group, as well as the use of GEEs to evaluate the data set which reduces data noise for interrelatedness of cells in repeated measures and the variability associated with heterogeneity of chronic pain. Further, use of comprehensive analyses to also consider tactile distance judgement task accuracy ensures our results of maintained anisotropic perception bias and general tactile anisotropy are not due to task error.

## CONCLUSION

In this study, we found no compelling evidence for disrupted spatial processing of tactile input on the hand dorsum in CRPS: tactile anisotropy and anisotropic perception bias were both comparable between CRPS patients, pain-controls and pain-free participants. The lack of a difference between groups in tactile anisotropy (and bias) supports the idea that S1 hand representation is differently disrupted than previously thought. Furthermore, the present findings suggest that the presumed relationship between tactile dysfunction,

S1 abnormalities and distorted hand perception should be revisited. Further research and clinical studies are required to interrogate the possible mechanisms that underpin hand perception distortion in CRPS.

## ACKNOWLEDGEMENTS

We thank Mrs. Aleisha Brock, Bachelor of Science in Statistics and Mathematics, research assistant and PhD Student at School of Nursing and Midwifery, University of South Australia. Mrs. Brock provided statistical consult regards the data analyses. We thank Dr. agr. Henrik Rudolf, Department of Medical Informatics, Biometry and Epidemiology, Ruhr-University Bochum for statistical attendance, analyses support and statistical supervision as well as support in crafting figures of result section. Finally, we are particularly grateful to the participants of this study who furthered our understanding the chronic pain by participating in this study.

### Funding

The authors received no funding for this work.

### Competing Interests

Lorimer Moseley receives royalties for books on pain education and rehabilitation and for professional development on pain education, in some cases focusing on CRPS. He also receives speaker fees for lectures on pain and rehabilitation. He leads the non-profit Pain Revolution Rural Outreach Tour and Local Pain Educators initiative. He is an Academic Editor for PeerJ.

Tasha Stanton received travel and accommodation support from Eli Lilly Ltd; this was unrelated to the present study.

### Author Contributions

- Annika Reinersmann conceived and designed the experiments, performed the experiments, analyzed the data, prepared figures and/or tables, authored or reviewed drafts of the paper, and approved the final draft.
- Ian W. Skinner, Thomas Lücke, Nicola Massy-Westropp and G. Lorimer Moseley conceived and designed the experiments, authored or reviewed drafts of the paper, and approved the final draft.
- Henrik Rudolf conceived and designed the experiments, prepared figures and/or tables, authored or reviewed drafts of the paper, and approved the final draft.
- Tasha R. Stanton conceived and designed the experiments, analyzed the data, prepared figures and/or tables, authored or reviewed drafts of the paper, and approved the final draft.

### Human Ethics

The following information was supplied relating to ethical approvals (i.e., approving body and any reference numbers):

This study was approved by the University of South Australia (UniSA) Human Research Ethics Committee (No. 35944). All recruited participants provided written informed consent and the study was performed in accordance with the ethical standards of the Declaration of Helsinki (1991).

## Data Availability

Raw data are available in the Supplemental Files.

## Supplemental Information

Supplemental information for this article can be found online at http://dx.doi.org/10.7717/peerj.11156#supplemental-information.

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
