# Peer review of "Intact tactile anisotropy despite altered hand perception in complex regional pain syndrome: rethinking the role of the primary sensory cortex in tactile and perceptual dysfunction"

_PeerJ, doi:10.7717/peerj.11156_

## Round 0.1 · original submission · Major Revisions

Two reviewers agree that the manuscript needs revision. Please revise and resubmit.

Reviewer 1 ·

Basic reporting

My main comments concern the terminology, which appears to be used somewhat inconsistently and in a confusing fashion, making it sometimes hard to follow the narrative of the paper. Table 1 very helpful in this regard, but seems to contradict some statements made in the paper. Specifically, ‘tactile sensitivity’ can be used generally, but in most cases refers to neural or perceptual thresholds regarding stimulation amplitude, at least in the psychophysical or neuroscientific literature that I am familiar with. ‘Tactile spatial acuity’ generally refers to tasks establishing the spatial resolution of the skin, such as two-points or grating orientation discrimination. I have not encountered the use of ‘tactile spatial acuity’ for tactile distance tasks.

Similarly, the paper seems to (correctly) suggest that tactile spatial acuity tasks using two-point discrimination paradigms can potentially be mastered using non-spatial cues. Tactile distance tasks are presented as a spatial alternative. I do not agree with this representation. Better versions of these tasks are grating orientation tasks, which also measure the same underlying variable, the tactile spatial resolution, however without non-spatial cues as confounds. Tactile distance judgements are inherently different tasks. They require an underlying spatial representation of the body from which tactile distances can be computed. Tasks that purely test spatial resolution on the other hand do not require such a representation. I apologize if I have misinterpreted the authors’ statements; this is how I interpreted the introduction. Nevertheless, I agree with the use of a tactile distance task as appropriate for this study.

Regarding the involvement of primary somatosensory cortex in tactile distance judgements, the authors might be interested in a recent preprint that contains new data on this question:
Tamè L, Tucciarelli R, Sadibolova R, Sereno MI, Longo MR. Reconstructing neural representations of tactile space. bioRxiv. 2019. doi:10.1101/679241

Experimental design

The experimental design is appropriate and well explained. Minor point: Perhaps ‘measure’ would be a better name for the four different ‘models’ introduced in the methods/results?

Validity of the findings

The paper fills an important gap in the literature and the results are intriguing. I agree very much with the authors’ statement that performance across groups needs to be tested across a multitude of perceptual assessments for a clear picture to emerge on the specific tactile processing changes in CRPS.

·

Basic reporting

See below

Experimental design

See below

Validity of the findings

See below

Additional comments

This paper, “Intact spatial acuity despite altered hand perception in complex regional pain syndrome: rethinking the role of the primary sensory cortex in tactile and perceptual dysfunction”, by Reinersmann and colleagues reports a study investigating tactile distance anisotropy in complex regional pain syndrome (CRPS). Patients with CRPS were compared to controls both with and without pain. Consistent with previous results, the authors report an anisotropy in tactile distance perception, with distances oriented across the width of the hand dorsum perceived as larger than those oriented along the length of the hand. No differences between groups were observed, however.

This study tests an interesting hypothesis about CRPS and appears to have been well conducted. Though no group differences are found, this null result is nevertheless notable and certainly worth documenting in the literature. I therefore believe that these data should be published. I do however have some comments and suggestions which I hope the authors find helpful in revising this paper.

My biggest comment is that the analyses are confusing and convoluted. While the experimental design seems to have been modelled closely on that used by Longo & Haggard (2011; JEP:HPP), the analysis is completely different, for no apparent reason that I could tell. Four separate “Models” of the data are presented, in addition to curve fitting. But I can’t see anything that these models add to just looking at the PSE and slope of the psychometric function. And Model 4 I couldn’t follow at all. Further, while it is claimed that there was significant anisotropy on both hands in all 3 groups, no actual statistical tests of this are reported.

As I wrote above, I think the only analysis needed is the curve-fitting one, but this is poorly described. No information is given about what kind of curve was fit, with what criteria of goodness-of-fit, how good the fit actually was, or which software was used. While the PSEs from these curves are presented in Figure 4, I can’t understand what is actually being plotted. Is the PSE the ratio between the across and along stimuli? If so, there does not in fact seem to be evidence for anisotropy. But I’m not sure that’s what these data are. It’s also unclear what the error bars depict (e.g., SEM, 95% CI, etc). Further, this seems to be an inappropriate use of a bar graph, since the y-axis is truncated at a seemingly random place (0.6).

Second, I can’t follow the logic of the power analysis (lines 113-118). This is based on an effect size of 0.2, but the authors don’t indicate which effect size measure this is supposed to be (e.g., Cohen’s d, partial eta-square, etc). Further, the analysis used is justified by “the repeated measures design”, but the whole point is to compare different groups of people, which obviously isn’t repeated measures. It isn’t clear which key statistical test this power analysis was based on, nor which software was used. I find it very difficult to see how 15 subjects per group can give 80% power for detecting any plausible smallest theoretically-meaningful effect size. I am sympathetic to the difficulty in finding patients, so do not think that the sample size should be a barrier to publication. But this power analysis is unconvincing and underspecified.

For the template matching task, I can’t understand how hand size was quantified. The “unmodified” hand image is a photo of somebody else’s hand, and not of the subject’s own. Further, subjects will differ from each other in terms of their actual hand size. So what are the statistical tests comparing?

Other points:
1. In the title, and elsewhere, the authors refer to their task as a measure of “tactile acuity”. While the slope of the psychometric function in a distance discrimination task could be considered a measure of acuity, for the most part what the authors are measuring here is BIAS, not acuity.
2. Line 16: It’s odd to say that the task “incorporates anisotropy”. The task measures anisotropy.
3. Line 18: It’s not correct that participants “judged the equivalence of tactile distances”. They in fact judged which one felt bigger.
4. Line 65: “Knight Fle et al” should be “Le Cornu Knight et al”.
5. Line 71: I don’t think it’s correct that Weber found that tactile distances across the limb are perceived as larger than those along the limb. He reported that tactile distances felt bigger on more sensitive skin surfaces and (separately) that 2-point discrimination thresholds were smaller across than along the limbs. But I don’t think he combined those two observations. I think that Green (1982, Perception & Psychophysics) is a better reference here than Weber.
6. Line 135: It would be good to add a citation or two for the template matching method used.
7. Line 163: It’s not true that there are only three possible distance combinations with this stimulus set. 4/3 is also possible.
8. Line 170-172: Do these authors really claim that tactile anisotropy is better measure of “S1 morphology” than acuity measures?
9. Line 175: What are the “further studies” referred to here?
10. Line 181: Did the subject judge whether the across or the along stimulus felt bigger? Or whether the first or the second stimulus felt bigger?
11. Line 187: There were 10 repetitions of each stimulus pair. But given that the stimuli were presented sequentially, was the order also counterbalanced?
12. For all the statistical analyses, appropriate measures of effect size would be a nice addition.

---

## Round 0.2 · accepted · Accept

Two reviewers have reviewed your study and feel you have addressed their concerns. Congratulations!

Reviewer 1 ·

Basic reporting

All my concerns have been addressed.

Experimental design

No comment.

Validity of the findings

No comment.

Additional comments

All my concerns have been addressed.

·

Basic reporting

See below

Experimental design

See below

Validity of the findings

See below

Additional comments

The authors have satisfactorily addressed the points I raised in my previous review. I support publication of the paper in its current form.